# A Randomized, Double-Blind, Placebo-Controlled Clinical Trial Assessing the Effects of *Angelica Gigas* Nakai Extract on Blood Triglycerides

**DOI:** 10.3390/nu12020377

**Published:** 2020-01-31

**Authors:** Su-Jin Jung, Woo-Rim Kim, Mi-Ra Oh, Youn-Soo Cha, Byung-Hyun Park, Soo-Wan Chae

**Affiliations:** 1Clinical Trial Center for Functional Foods, Chonbuk National University Hospital, Jeonju, Jeonbuk 54907, Korea; sjjeong@jbctc.org (S.-J.J.); wrkim@jbctc.org (W.-R.K.); mroh@jbctc.org (M.-R.O.); 2Biomedical Research Institute, Chonbuk National University Hospital, Jeonju, Jeonbuk 54907, Korea; 3Department of Food Science and Human Nutrition, Chonbuk National University, Jeonju, Jeonbuk 54896, Korea; cha8@jbnu.ac.kr; 4Department of Biochemistry and Molecular Biology, Chonbuk National University Medical School Jeonju, Jeonbuk 54896, Korea; bhpark@jbnu.ac.kr; 5Department of Pharmacology, Chonbuk National University Medical School, Jeonju, Jeonbuk 54896, Korea

**Keywords:** *Angelica gigas* Nakai, triglycerides, VLDL-C, TG/HDL-C, atherogenic index

## Abstract

*Angelica gigas* Nakai, Korean dang-gui, has long been widely used in traditional treatment methods. There have been a number of studies of the health effects of *A. gigas* and related compounds, but studies addressing effects on blood triglycerides (TG) are lacking. To investigate the effects of *A. gigas* Nakai extract (AGNE) on TG in Korean subjects, we carried out a 12-week, randomized, double-blind, placebo-controlled clinical trial. Subjects who met the inclusion criterion (130 mg/dL ≤ fasting blood TG ≤ 200 mg/dL) were recruited for this study. One hundred subjects were assigned to the AGNE group (*n* = 50) or the placebo group (*n* = 50), who were given 1 g/day of AGNE (as *a gigas* Nakai extract 200 mg/d) in capsules and the control group for 12 weeks. Outcomes were efficacy TG, lipid profiles, atherogenic index, and safety parameters were assessed initially for a baseline measurement and after 12 weeks. After 12 weeks of supplementation, TG and very low-density lipoprotein cholesterol (VLDL-C) concentration and TG/HDL-C ratio in the AGNE group were significantly reduced compared to the placebo group (*p* < 05). No significant changes in any safety parameter were observed. These results suggest that the ingestion of AGNE may improve TG and be useful to manage or prevent hypertriglyceridemia.

## 1. Introduction

The prevalence of metabolic disease is increasing dramatically due to changes in diet and lifestyle over the last few decades. According to the 2016 Korean National Health and Nutrition Examination Survey, the average prevalence of hypertriglyceridemia is 17.5 % [1]. Hypertriglyceridemia is usually treated using lipid lowering agents, such as fibrate, niacin, bile acid-binding resin, and statins [2,3]. However, these treatments are reported to be accompanied by numerous side effects, such as diarrhea, severe muscle damage, dry skin, and liver dysfunction [4,5,6]. These side effects have led a growing number of people to seek natural remedies, including functional foods and dietary choices, that may help to improve the lipid profile with minimal side effects [7]. These days, people are highly interested in healthy living and in the prevention and treatment of chronic diseases; as a result of this, research on physiologically active and functional foods is on the rise [8,9,10]. *Angelica* species (dang-gui) are native to north-eastern areas of China and Korea. Korea dang-gui is also called ‘cham-dang-gui’ and grows at high altitude in cool regions in Korea. Dang-gui has long been widely used for treatment of gynecological diseases, improvement of blood flow [11], and diabetic hypertension [12] and for its anti-inflammatory [13] and anti-cancer activities [14,15], anti-diabetic activity in streptozotocin-induced diabetic mice [16], and anti-oxidant activity in scavenging DPPH (1,1-diphenyl-2-picrylhydrazyl) radicals [17]. We recently conducted experiments to determine whether AGNE can alleviate dyslipidemia in mice fed a high fat diet (HFD). Body weight and blood glucose and insulin concentrations were reduced in AGNE-treated mice. We also found dose-dependent effects of AGNE on serum and hepatic TG concentrations [18]. We therefore undertook a placebo-controlled clinical trial of AGNE in the treatment of dyslipidemia. Here, we report the effects of AGNE supplementation for 12 weeks on blood TG, other lipid profiles, and the atherogenic index.

## 2. Materials and Methods

### 2.1. Design

This study was a 12-week, randomized, double-blind, placebo-controlled parallel group clinical trial performed according to a computer-generated randomization schedule designed to assign subjects to the ANGE or placebo groups. A total of 100 subjects were randomly assigned (1:1) to one of the study groups (50 subjects each) using a computer-generated random number table generated by the randomization program of SAS^®^, version 9.4 (SAS Institute, Cary, NC, USA).

### 2.2. Subjects 

The study subjects were recruited and selected from the Clinical Trials Center for Functional Foods (CTCF2) at Chonbuk National University Hospital from March 2017 to April 2018. All subjects gave written informed consent before entering the study. The Helsinki Declaration guidelines were applied in this study, and the Chonbuk National University Hospital (CUH) Institutional Review Board (IRB) for Functional Foods approved all proceedings (approval No.: 2016-02-032); the approval was subsequently transferred to the CUH IRB. This clinical trial was registered at Clinical Trials.gov (www.clinicaltrials.gov) under NCT03079648. The criteria for selection and exclusion of participants in this study are described below.

Selection criteria: 1)Adult male and female participants aged 19–80 years at the time of screening2)Participants with a blood TG range of 130~200 mg/dL at screening.3)Participants who fully understood the test and decided to participate of their own free will and agreed to the written consent document.

Exclusion criteria: 1)Subjects who were treated with a lipid-lowering agent within six months of undergoing treatment2)History of severe CVD, such as heart attack and/or stroke3)History of genetic hyperlipidemia and kidney diseases, such as acute or chronic renal failure; autoimmune disease, cancer, respiratory disease (asthma, chronic obstructive pulmonary disease); or diagnosis of diabetes4)History of clinically significant hypersensitivity reaction5)History of gastrointestinal disease (e.g., Crohn’s disease) or surgery (excluding appendectomy and herniotomy)6)Antipsychotic therapy within two months prior to screening7)History of alcoholism or drug abuse8)Participation in other clinical trials within two months prior to screening9)Aspartate aminotransferase (AST) and alanine aminotransferase (ALT) concentrations over three times the upper limit of the reference range or serum creatinine > 2.0 mg/dL10)Pregnant or nursing women11)No use of appropriate birth control methods among fertile women (except surgery for female infertility)12)Deemed unfit by the principle investigator due to other complications.

### 2.3. Test Supplements

In our earlier study, we found that TG concentration was reduced after ANGE supplementation in the high fat diet mouse model, and that the greatest improvement in TG concentration was noted in mice receiving ANGE 40 mg/kg [18]. Based on the results, the appropriate ANGE dose for subjects in the present study was determined to be 200 mg/day. Test supplements were developed by Wellness Up Research Inc. (Anyang, Gyeonggi, Republic of Korea) [19] for use in a mouse study. Placebo capsules were prepared without any active components of the ANGE supplement, and the ingredients were considered harmless to humans and were judged to have no effect on blood TG. Subjects received either the ANGE or placebo capsules every week, and all of the subjects were instructed to take either one ANGE capsules (twice per day) or one placebo capsules (twice per day) per day (1 g/day) after breakfast and dinner for 12 weeks. The analytical results if the test and placebo supplements are provided in Table 1.

### 2.4. Standard Meal

Several studies have shown that blood TG concentration is affected by diet, drinking, exercise, and drugs [20,21]. Therefore, to minimize potential confounding in this study, all subjects visited the food court in the CUH the day before screening and on their third visit and consumed a standard dinner-time meal [beef bulgogi (Korean beef stew) with rice; total 640 kcal]. The subjects fasted for 12 h after the standard meal and were strictly advised not to consume any food, including water during this time. Detailed nutritional information for the standard diet was analyzed using CAN-Pro 4.0^®^ software (The Korean Nutrition Society, Seoul, Korea) as summarized in Appendix A.

### 2.5. Outcome Measurements

The efficacy and safety evaluation parameters at baseline and after 12 weeks of assigned diet were analyzed prior to participation in this study.

The primary outcome was change in blood TG concentration after 12 weeks of supplementation with AGNE. The secondary outcomes were changes in other lipid parameters (total cholesterol [TC], LDL-C, high-density lipoprotein cholesterol [HDL-C], non-HDL-C, very low-density lipoprotein cholesterol [VLDL-C], free fatty acid, apolipoprotein A1 [apoA1], and apoB, high-sensitivity C reactive protein (hs-CRP), and atherogenic index (TC/HDL-C, LDL-C/HDL-C, TG/HDL-C, [TC–HDL-C]/HDL-C, apoB/apoA1) after 12 weeks of the assigned diet. Non-HDL-C, VLDL-C, and LDL-C were calculated using the equation.

### 2.6. Safety Outcome Measurements

Safety outcomes were assessed by recording the incidence of serious or minor adverse events, laboratory test results, and vital signs (systolic [SBP] and diastolic [DBP] blood pressure and pulse rate). A laboratory profiles, was also conducted. Blood samples were collected after a 12-h fast and kept frozen at −80 °C until analysis after performing centrifugation (Hanil Science Industrial Co. Ltd., Seoul, Korea) at 3000 rpm for 20 min. Hematological parameters assessed were WBC, RBC, hemoglobin, hematocrit, and platelet count. Blood biochemical tests were conducted to assess total bilirubin, ALP, gamma-GT, ALT, AST, total protein, albumin, BUN, creatinine, glucose, creatinine kinase, and LDH. Urine samples were examined for specific gravity, pH, WBC, nitrite, protein, ketone, bilirubin, urobilinogen, and occult blood. All biochemical analyses were performed by the clinical pathology department of our hospital. 

### 2.7. Evaluation of Diet and Physical Activity

During the intervention period of 12 weeks, subjects were asked to continue their usual diets and activity and to not ingest any other functional foods or dietary supplements. Dietary intakes were monitored by a registered dietician according to information obtained from each visit to evaluate subject’s three-day usual diet. The dietary intake analysis was performed using Can-pro 4.0 (Computer aided nutritional analysis program, The Korean Nutrition Society Forum, Seoul, Korea) with the data from the 3 days recorded during the 12 weeks study from which the average daily calorie and nutrient intake were calculated. Physical activity was evaluated according to a metabolic equivalent task (MET) assessment using the global physical activity questionnaire (GPAQ). The MET value was used for analysis of physical activity or GPAQ data, representing the relative proportion of working metabolic rate to metabolic rate at rest.

### 2.8. Statistical Analysis

All statistical analyses were performed using SAS^®^ version 9.4 (SAS Institute, Cary, NC, USA). The primary and secondary outcome analyses were based on per protocol set (PPS) analysis of subjects who completed the clinical trial in accordance with the protocol. Safety analysis was performed on safety subjects who received at least one treatment. Values were expressed as the mean ± SD (standard deviation). Differences between intake groups were analyzed using an independent *t*-test, and differences in variation within the intake group were analyzed using a paired *t*-test. Categorical variables were analyzed using the chi-square test. Analysis of variance (ANCOVA) adjusted for MET value variations was used to determine whether the observed difference in blood TG concentration between the two groups was independent of MET value [22,23]. Also the outcome variables for repeated measurement of the intake groups were applied with a linear mixed model between the intake groups. The significance was statistically significant at the level of *p* < 0.05. 

### 2.9. Sample Size

The sample size calculation for this study was based on the assumption of −20.4 mg/dL variation in measured blood TG concentration after 12 weeks of AGNE supplementation, −5.3 mg/dL variation in the placebo group, and standard deviation of 27.3 mg/dL. The number of subjects required was calculated as described in Suliburska et al. (2012) [24]. The number of subjects needed in each group to achieve an 80% power for a 5% significance level with a two-sided test was 40; therefore, 100 subjects were enrolled for 1:1 randomization to the AGNE or placebo group.

## 3. Results

### 3.1. Participant Demographic Characteristics

Four subjects withdrew consent during the study, and one subject with serious adverse effects and another using a prohibited drug were eliminated from the study, and one subject was excluded from the analysis because of an unmeasured efficacy factor (apolipoprotein B, apo B). Therefore, a total of 93 subjects (AGNE group: 48 subjects, placebo group: 45 subjects), which exceeded the target number of subjects (80 subjects), completed all procedures specified in the protocol (Figure 1). The demographic information for the study subjects is summarized in Table 2. There were no statistically significant differences between the groups with regard to demographic characteristics. The compliance rates, which were based on capsule count, were 94.6 ± 6.6% and 93.4 ± 6.4% in the placebo and AGNE groups, respectively (*p* > 0.05).

### 3.2. Diet Intake and Physical Activity

When dietary intake was analyzed for per protocol set (PPS) of efficacy analysis performed, there were no statistically significant difference between groups (Table 3). However, there was a statistically significant difference in physical activity (MET value) between groups, with decreased physical activity in AGNE group and increased in placebo group (*p* < 0.05).

### 3.3. Efficacy Evaluation

#### 3.3.1. Primary Outcome

Table 4 shows the blood TG concentration measured at 12 weeks. A significant reduction in TG concentration was noted in the AGNE group (absolute change: −14.8 ± 16.1 mg/dL) compared to baseline (*p* = 0.010), whereas no change was evident in the placebo group (absolute change: +19.9 ± 66.4 mg/dL, *p* = 0.131). The difference between groups was statistically significant including physical activity adjusted analysis (*p* < 0.05).

#### 3.3.2. Secondary Outcome

Table 4 also presents the blood lipid profiles at 12 weeks. Significant reductions in VLDL-C concentration in the AGNE group (*p* = 0.010) and free fatty acids in the placebo group (*p* = 0.025) were observed compared to baseline. In addition, TG/HDL-C ratio was reduced in the AGNE group (*p* = 0.013) compared to baseline. The difference of TG, free fatty acid, VLDL-C concentrations, and TG/HDL-C ratio between groups was statistically significant after 12 weeks supplementation with AGNE compared to the placebo group including physical activity adjusted analysis (*p* < 0.05). There were no statistically significant differences between the groups for any other variables.

### 3.4. Safety 

Among the 100 subjects in the clinical trial, 27 (AGNE group: 17 cases, placebo group: 10 cases) experienced mild or moderate adverse events, and 20 experienced a serious adverse event during the study period. A total of 17 cases of adverse events in AGNE group have reported upper respiratory tract infection (4), indigestion (1), bloating (1), runny nose (1), hordeolum (1), liver enzyme elevation. (2), epigastric discomfort (1), lumbago (1), dizziness (1), gingivitis (1), headache (1), nasal hemorrhage (1), and dry eye (1). Also, a total of 10 cases of adverse events in placebo group have reported upper respiratory tract infection (2), allergic rhinitis (1), gingivitis (1), omalgia (1), coxalgia (1), diarrhea (1), chill (1), myalgia (1), and nausea (1). AGNE group experienced one case of thyroid cancer of serious adverse event. Regarding the serious adverse events, the study team judged that there was no causal relationship with the treatment, and the CUH IRB concurred. After analysis of the adverse events, there were no significant differences between the occurrence and types of adverse event between the groups (data not shown). When analyzing the results of laboratory testing (blood and urinalysis), ECG, vital signs, and anthropometric evaluations, no significant statistical or clinical differences between the groups were noted (Appendix A).

## 4. Discussion

Hyperlipidemia is one of the leading causes of early death and is expected to become the most important cause of death in developed and developing countries [25]. Hyperlipidemic patients are likely to develop arteriosclerosis and blood clots and may also develop unstable angina or myocardial infarction [26]. For this reason, the main purpose of this study was to assess the effects of AGNE supplementation on TG and blood lipid concentrations. The present study found that TG, VLDL-C, and free fatty acid concentrations and TG/HDL-C ratio improved after 12 weeks supplementation with AGNE compared to the placebo group. Several previous studies have reported a negative correlation between plasma TG concentration and physical activity [22,25,27]. Interestingly, the subjects in the AGNE group reported significant reduction in physical activity, but TG concentration was reduced. This result suggests that there is another pathway along which AGNE suppresses TG level regardless of physical activity.

The efficacy evaluation in this study confirmed that blood TG and VLDL-C levels were significantly reduced in the AGNE group compared to the placebo group. Our previous animal study found similar results, in that AGNE supplementation for 16 weeks reduced TG concentrations in plasma and liver. Further mechanistic studies in HepG2 cells indicated that AGNE increased expression of carnitine palmitoyltransferase-1 (CPT1) and peroxisome proliferator-activated receptor α (PPARα) and suppressed expression of p-acetyl CoA carboxylase (ACC), fatty acid synthase (FAS), CD36, stearoyl-CoA desaturase-1 (SCD-1), and SREBP-1. These effects were mediated by the activation of the Sirt1-adenosine monophosphate-activated protein kinase (AMPK) pathway [18]. In preclinical study [18], we observed that AGNE supplementation decreased serum TG and TC levels compared with the HFD group. However, there was no significant difference in HDL-C and LDL-C levels between the HFD groups. Our findings are similar to previous animal studies demonstrating hyperlipidemia effects of AGNE. During this study, there was no difference in TC, LDL-C and HDL-C levels between the AGNE and placebo groups. It is believed that the mechanism mediate that improved the hypertriglyceridemia in these groups was the significant. Because of supplementation of AGNE, which contains of decursin and decursinol angelate, it suggests that may have helped not only suppress lipogenesis by activating Sirt1-AMPK signaling but also reduce TG level by accelerating β oxidation of fatty acids. We also found that AGNE supplementation increased insulin sensitivity in *db/db* mice and consequently prevented conversion of blood glucose into TG in the liver [28]. Together, these findings indicate that AGNE reduces hepatic and plasma TG concentrations by suppressing de novo lipogenesis and accelerating β oxidation of fatty acids. The TG/HDL-C ratio is one component of the atherogenic index and improved in the AGNE group in this study. One of the most important factors in cardiovascular disease (CVD) is hyperlipidemia. Reduction in HDL-C and increases in TC, LDL-C, and TG contribute to development of atherosclerosis [29,30]. Similarly, a study by Cai et al (2017) indicated that the atherogenic index is the strongest predictor of CVD, and TG has the highest positive correlation with the atherogenic index [31]. Several studies have reported that the risks of myocardial infarction and myocardial ischemia increase when TG/HDL-C ratio increases, which may ultimately lead to CVD [32]. Given that the TG/HDL-C ratio is a strong predictor of heart disease [33,34], we concluded that the reduction in TG/HDL-C ratio in this study was a very positive result. Analysis of safety indicators following AGNE supplementation indicated that any changes were within the normal range and had no clinical significance. Therefore, we consider short-term AGNE supplementation to have been safe in our study population. The strength of this study was control of diet prior to testing by introducing a standard diet on the day before screening and the day before the end of the 12-week visit to minimize dietary factors that could affect the concentration of fasting triglycerides in the blood. A more sophisticated assessment was made by minimizing the deviation of TG, which is an indicator of effectiveness of the study. However, there are some limitations to this study that should be considered. First, this study included a small number of participants; therefore, caution should be used in generalizing the results to other populations with hyperlipidemia. Second, the changes in TC, LDL-C, and HDL-C observed in hyperlipidemia subjects treated with AGNE supplementation was not significant. Therefore, a long-term intake period would be required to verify the efficacy of the AGNE. 

## 5. Conclusions

We found that blood TG and VLDL-C concentrations and TG/HDL-C ratio were reduced after consuming AGNE for 12 weeks compared to placebo. During the study, no clinically meaningful adverse events or alterations in laboratory test results (blood and urinalysis), ECG, vital signs, or anthropometric evaluations were evident, confirming the safety of AGNE in our study population. Therefore, we conclude that AGNE has positive effects when used as a drug substitute to treat dyslipidemia, and we feel that our study represents a meaningful contribution to research focusing on the functional role of *A. gigas* Nakai in promoting human health.

## Figures and Tables

**Figure 1 nutrients-12-00377-f001:**
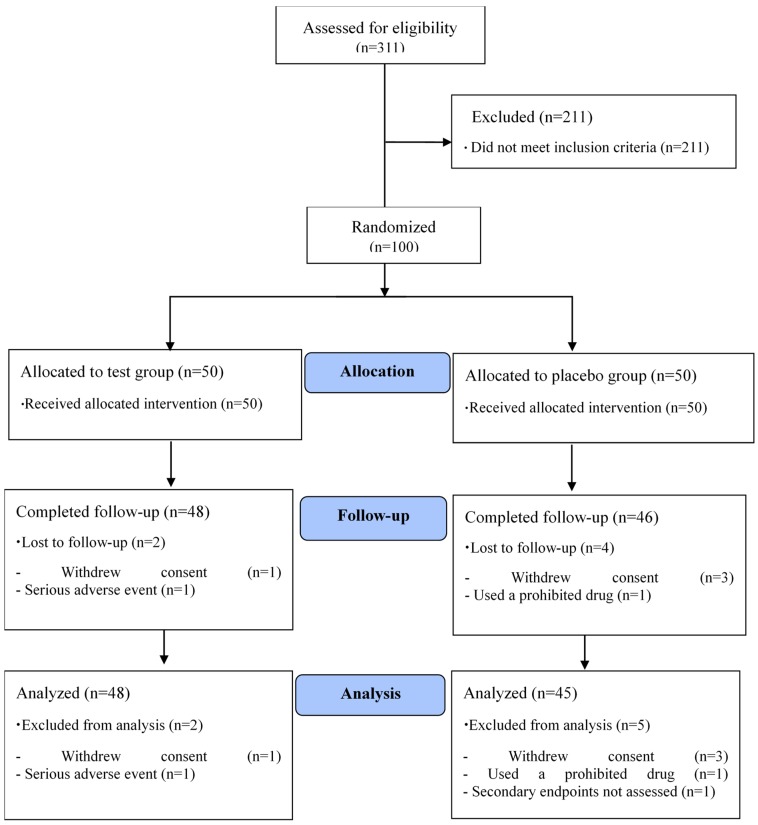
Flow diagram of the participants in this human study.

**Table 1 nutrients-12-00377-t001:** Composition of the test product as determined by high-performance liquid chromatography (HLPC).

Component	Test Capsule *A. Gigas* Nakai Extract Supplement (%)	Placebo Supplement (%)
*A. gigas Nakai* extract	20	-
Soybean oil	73	93
Beeswax	7	7
Total	100	100

**Table 2 nutrients-12-00377-t002:** Baseline general characteristics of the subjects.

	AGNE Group (*n* = 50)	Placebo Group (*n* = 50)	Total (*n* = 100)	*P* Value ^1)^
Sex (M/F)	17/33	16/34	33/67	0.832 ^2)^
Age (years)	49.1 ± 10.5	50.5 ± 10.3	49.8 ± 10.4	0.497
Height (cm)	162.4 ± 8.1	163.2 ± 9.4	162.8 ± 8.8	0.634
Weight (kg)	67.9 ± 11.9	69.1 ± 14.3	68.5 ± 13.1	0.631
BMI (kg/m^2^)	25.6 ± 3.3	25.8 ± 3.6	25.7 ± 3.4	0.809
SBP (mmHg)	124.2 ± 15.7	123.8 ± 13.0	124.0 ± 14.6	0.889
DBP (mmHg)	78.4 ± 11.9	78.0 ± 10.0	78.2 ± 10.9	0.856
Pulse (bpm)	77.2 ± 8.3	74.0 ± 11.3	76.7 ± 9.5	0.631
Alcohol (Y/N)	21/29	18/32	39/61	0.539 ^2)^
Alcohol (unit ^3)^/week)	4.1 ± 4.7	4.1 ± 5.7	4.1 ± 5.1	0.966
Smoking (Y/N)	5/45	5/45	10/90	>0.999 ^2)^
Smoking (n/day)	9.2 ± 7.1	16.0 ± 5.5	12.6 ± 7.0	0.129

Values are presented as mean ± SD or number; ^1)^ Analyzed by independent t test; ^2)^ Analyzed by chi-square test; ^3)^ Alcohol 1 unit = Alcohol 10 g = Alcohol 12.5 mL; Abbreviations: BMI, Body mass index; SBP, Systolic blood pressure; DBP, Diastolic blood pressure.

**Table 3 nutrients-12-00377-t003:** Major nutrient intakes and metabolic equivalent values of the AGNE and placebo groups measured at baseline and 12 weeks.

	AGNE Group (*n* = 48)				Placebo Group (*n* = 45)				
	Baseline	Week 12	Change	*P* Value ^1)^	Baseline	Week 12	Change	*P* Value ^1)^	*P* Value ^2)^
Energy (Kcal)	1595.3 ± 466.4	1627.0 ± 476.0	31.7 ± 10.0	0.662	1600.2 ± 399.2	1591.9 ± 428.9	−8.3 ± 29.7	0.885	0.667
Carbohydrates (g)	249.3 ± 73.0	240.4 ± 68.1	−8.9 ± 65.5	0.351	243.0 ± 68.4	249.1 ± 71.2	6.1 ± 2.8	0.549	0.281
Lipids (g)	38.4 ± 20.2	45.2 ± 23.3	6.8 ± 3.15	0.102	40.4 ± 15.6	39.1 ± 18.7	−1.3 ± 3.1	0.677	0.117
Protein (g)	62.1 ± 19.5	63.8 ± 20.6	1.7 ± 1.1	0.582	62.5 ± 15.9	60.1 ± 17.8	−2.4 ± 1.9	0.366	0.314
Fiber (g)	21.0 ± 8.6	20.1 ± 7.2	−0.9 ± 1.4	0.377	19.8 ± 7.0	20.0 ± 7.1	0.3 ± 0.1	0.787	0.409
MET value(min/week)	2383.8 ± 3198.4	1490.0 ± 1840.1	−893.8 ± 1358.3	0.051	1351.1 ± 2190.3	1947.6 ± 2779.3	596.4 ± 589	0.139	0.015

Values are presented as mean ± SD; ^1)^ Analyzed by paired *t* test; ^2)^ Analyzed by a linear mixed model (differences between groups). Abbreviation: MET, metabolic equivalent.

**Table 4 nutrients-12-00377-t004:** Changes in blood triglycerides, lipid profile, high sensitivity C-reactive protein, and atherogenic index.

	AGNE Group (*n* = 48)				Placebo Group (*n* = 45)					
	Baseline	Week 12	Change	*P*-Value ^1)^	Baseline	Week 12	Change	*P*-Value ^1)^	*P*-Value ^2)^	*P*-Value ^3)^
TG (mg/dL)	158.2 ± 21.3	143.4 ± 37.4	−14.8 ± 16.1	0.010	159.4 ± 20.3	179.4 ± 86.7	19.9 ± 66.4	0.131	0.013	0.009
TC (mg/dL)	213.1 ± 32.1	213.6 ± 29.7	0.5 ± 2.4	0.902	213.8 ± 34.6	209.3 ± 34.3	−4.6± 0.3	0.330	0.403	0.527
LDL-C (mg/dL)	135.8 ± 26.7	133.5 ± 25.6	− 2.3 ± 1.1	0.493	134.4 ± 30.3	126.1 ± 29.7	−8.4 ± 24.2	0.025	0.218	0.304
HDL-C (mg/dL)	50.4 ± 8.5	51.4 ± 8.0	1.1 ± 0.5	0.178	50.6 ± 11.2	50.3 ± 11.8	−0.29 ± 0.6	0.802	0.326	0.617
Non-HDL-C (mg/dL)	162.8 ± 27.8	162.2 ± 27.1	−0.6 ± 0.7	0.875	163.2 ± 30.1	158.9 ± 30.1	−4.3 ± 0.1	0.330	0.516	0.594
VLDL-C (mg/dL)	31.6 ± 4.3	28.7 ± 7.5	−3.0 ± 3.2	0.010	31.9 ± 4.1	35.9 ± 17.3	4.0 ± 13.2	0.131	0.013	0.009
Free fatty acid (uEq/L)	596.1 ± 228.2	629.6 ± 209.7	33.5 ± 18.5	0.218	634.0 ± 219.5	572.1 ± 164.5	−62.0 ± 55.0	0.025	0.013	0.033
ApoA1 (g/L)	1.45 ± 0.19	1.47 ± 0.17	0.01 ± 0.02	0.546	1.47 ± 0.23	1.46 ± 0.21	−0.01 ± 0.02	0.707	0.498	0.423
ApoB (g/L)	1.21 ± 0.21	1.20 ± 0.21	−0.01± 0.0	0.723	1.22 ± 0.23	1.1 ± 0.23	−0.05 ±0.0	0.092	0.290	0.567
hs-CRP (g/L)	1.19 ± 1.81	1.20 ± 2.58	0.01 ± 0.8	0.984	0.57 ± 0.98	1.41 ± 5.18	0.8 ± 4.2	0.264	0.327	0.328
TC/HDL-C	4.28 ± 0.60	4.21 ± 0.62	−0.08 ± 0.02	0.349	4.34 ± 0.77	4.30 ± 0.86	−0.0 ± 0.09	0.640	0.812	0.710
LDL-C/HDL-C	2.74 ± 0.56	2.64 ± 0.56	−0.10 ± 0.0	0.132	2.75 ± 0.73	2.61 ± 0.78	−0.1 ± 0.05	0.060	0.658	0.497
TG/HDL-C	3.23 ± 0.70	2.87 ± 0.92	−0.36 ± 0.03	0.013	3.34 ± 0.99	3.87 ± 2.23	0.5 ± 1.24	0.074	0.006	0.006
(TC-HDL-C)/HDL-C	3.28 ± 0.60	3.21 ± 0.62	−0.08 ± 0.02	0.349	3.3 ± 0.77	3.3 ± 0.86	− 0.0 ±0.09	0.640	0.812	0.710
ApoB/ApoA1	0.84 ± 0.18	0.83 ± 0.16	−0.02 ± 0.02	0.410	0.85 ± 0.19	0.82 ± 0.20	−0.03 ± 0.01	0.131	0.597	0.646

Values are presented as mean ± SD; ^1)^ Analyzed by paired *t* test; ^2)^ Analyzed by a linear mixed model (differences between groups); ^3)^ Analyzed by ANCOVA, which was adjusted for MET value with change between baseline and 12 weeks. Abbreviations: MET, metabolic equivalents value; TG, triglyceride; TC, Total cholesterol; HDL-C, High-density lipoprotein cholesterol; LDL-C, Low-density lipoprotein cholesterol; VLDL-C, Very low-density lipoprotein cholesterol; hs-CRP, high sensitivity C-reactive protein.

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
