# Peer review of "A Randomized, Double-Blind, Placebo-Controlled Clinical Trial Assessing the Effects of Angelica Gigas Nakai Extract on Blood Triglycerides"

_nutrients, 2020, doi:10.3390/nu12020377_

Round 1
Reviewer 1 Report
The manuscript entitled: “A randomized, double-blind, placebo-controlled clinical trial. Assessing the effects of Angelica gigas Nakai extracts on blood triglycerides” is a well-designed study evaluating potential values of the plant extract (AGNE) in prevention of hypertriglyceridemia.
-The presented clinical trial is supported by the mouse model data published in the Journal of Functional Foods 2017, 31: 208-216.
-The introduction is coherent and well brings to the topic of the paper, giving a brief summary about already identified effects exerted by Angelica species. The paragraph contains a number of valuable information well summarizing previous and current studies, presenting also the gaps in the current state-of-the-art.
-Description of the ‘materials and methods” is of good quality, with sufficient amount of details.
-The data are appropriately presented and analysed, with proper statistical analysis.
-The conclusions reached, are consistent with the presented data.
-The content of the article well matches to the profile of the journal.
Author Response
Dear reviewer 1,
Thank you for your careful review of our manuscript.
Reviewer 2 Report
This is a randomized, double-blind, placebo-controlled clinical trial to evaluate AGNE on TG in Koran subjects. Overall, this is a fair study design, the authors conducted a standard clinical trial and appropriate statistics and considered the detailed procedure. They found AGNE could improve TG and suggested to management or prevent hypertriglyceridemia. Some suggestions may improve the study more completely. 1. Suggest to present serious or minor adverse events more detail. What kind of adverse events in this study? Figure 1 showed 1 subject had a serious adverse event. However, there is no definition of the safety outcome measurement in methods (page 5). Besides, what kind of tests included in the laboratory examination? 2. The efficacy in this study is only TG but no other lipid profiles. Suggest to discuss why not other lipid profiles have no influence by AGNE. 3. Suggest to add another paragraph to discuss the pros and cons about this study.Author Response
Dear Reviewer 2,
Reviewer reports:
Some suggestions may improve the study more completely.
A1. Suggest to present serious or minor adverse events more detail. What kind of adverse events in this study? Figure 1 showed 1 subject had a serious adverse event. However, there is no definition of the safety outcome measurement in methods (page 5). Besides, what kind of tests included in the laboratory examination?
Response: As you suggested, revised as suggested.
1) Adverse events: We made following changes (Line 6-12, page 13)
- Figure 1 showed 1 subject had a serious adverse:
=> AGNE group experienced 1 case of thyroid cancer of serious adverse event.
2) Laboratory examination is described in detail
(Line 26-30, page 5)~(Line 1-4, page 6)
A2. The efficacy in this study is only TG but no other lipid profiles. Suggest to discuss why not other lipid profiles have no influence by AGNE.
Response: As you suggested, revised as suggested (Line 9-18, page 14).
A3. Suggest to add another paragraph to discuss the pros and cons about this study.
Response: As you suggested, revised as suggested (Line 3-12, page 15).
Thank you!
